# Understanding political communication and political communicators on Twitch

Sangyeon Kim [1,2]*

1 Observatory on Social Media, Indiana University, Bloomington, IN, United States of America, 2 Luddy School of Informatics, Computing, and Engineering, Indiana University, Bloomington, IN, United States of America

* ski15@iu.edu

## Abstract

As new technologies rapidly reshape patterns of political communication, platforms like Twitch are transforming how people consume political information. This entertainment-oriented live streaming platform allows us to observe the impact of technologies such as "live-streaming" and "streaming-chat" on political communication. Despite its entertainment focus, Twitch hosts a variety of political actors, including politicians and pundits. This study explores Twitch politics by addressing three main questions: 1) Who are the political Twitch streamers? 2) What content is covered in political streams? 3) How do audiences of political streams interact with each other? To identify political streamers, I leveraged the Twitch API and supervised machine-learning techniques, identifying 574 political streamers. I used topic modeling to analyze the content of political streams, revealing seven broad categories of political topics and a unique pattern of communication involving context-specific "emotes." Additionally, I created user-reference networks to examine interaction patterns, finding that a small number of users dominate the communication network. This research contributes to our understanding of how new social media technologies influence political communication, particularly among younger audiences.

## 1 Introduction

As new technologies in the social media environment are rapidly reshaping patterns of political communication [1], how people consume political information via social media changes dramatically. Twitch, the entertainment-oriented live streaming platform, provides a novel environment for observing how "live-streaming" and "streaming-chat" technologies are influencing political discourse. Although the integration of politics into entertainment and apolitical spaces is not a new phenomenon [2–6], Twitch provides a distinctive case due to its highly interactive and real-time communication features. Notable political figures such as Alexandria Ocasio-Cortez have utilized Twitch to engage with audiences, streaming on the platform multiple times since 2020.

The popularity of political streamers such as Hasanabi, whose live coverage of the 2020 U.S. presidential election attracted significant attention, underscores the growing role of Twitch in political communication [7]. This trend raises important questions about why people

**Data Availability Statement:** Data related to the paper is all available in the following dataset archive of Harvard Dataverse: https://doi.org/10.7910/DVN/AE86N1.

**Funding:** This work was supported in part by Swiss National Science Foundation (SNSF Grant

#209250). The funders had no role in study design, data collection and analysis, decision to publish, or preparation of the manuscript.

**Competing interests:** The authors have declared that no competing interests exist.

increasingly consume political content from individual broadcasters on platforms like Twitch, rather than from traditional news media outlets.

Twitch's unique technological features, including real-time interaction, video and audio content, and the streaming chat function, establish a different dynamic between content creators and their audiences. These features grant streamers credibility and influence, as they can directly engage with their audience in real time [8–10]. Viewers, in turn, often perceive streamers as more relatable and accountable than political figures on other social media platforms [11]. Additionally, Twitch streamers rely heavily on audience interaction for both content creation and financial support, creating a feedback loop that emphasizes the audience's role in shaping political discourse on the platform [12]. While the spread of political speech into streaming media is not new [1], Twitch's technological affordances create a distinctive environment for these interactions.

The rise in political activities on Twitch, even though it is primarily an entertainment platform, calls for more systematic research into the dynamics of Twitch politics. While politics has long found its way into entertainment spaces [2–4, 6], the distinctive real-time and interactive nature of Twitch presents a fresh opportunity to study how these technological affordances shape political communication. On Twitch, political streamers engage with issues ranging from Black Lives Matter [13, 14] to the Capitol riots [15], often in ways that differ from traditional media formats.

Another significant aspect of Twitch is its user demographic, which skews younger compared to other major social media platforms, such as Twitter and Facebook. While more than 70% of users fall into the 16–34 age group, there is still some presence of older users, with 17% between the ages of 35–44, and 10% above 45 [16]. The predominance of younger users on Twitch offers a unique context for examining how political communication occurs among different age groups, particularly as younger generations might engage with political content differently than older generations. This generational composition may provide additional insights into how political discourse is shaped and consumed within these spaces, especially in comparison to platforms with broader age distributions.

To study the patterns of political communication in Twitch which has been understudied despite its theoretical and practical significance, I aim to answer three questions on Twitch politics in this paper by using various computational methods.

1. *Who are political Twitch streamers?* As Twitch is an extremely entertainment-oriented platform, the identification of political actors is essential to study Twitch politics. I found 574 political streamers using Twitch API and supervised machine learning techniques.

2. *What types of political content are discussed?* Once political streamers are identified, it becomes important to analyze the political content they cover. By collecting chat posts of political streamers and fitting topic models, I found political topics covered in the platform can be categorized into seven broad categories and a Twitch-specific pattern of political communication—the usage of context-specific "emotes (Twitch version of emoji)".

3. *How do the political streamers and their audiences interact with each other?* Twitch's live streaming and chat functions enable frequent interaction between streamers and their audiences. By analyzing user-reference networks within political streamers' chatrooms, I found that these networks often exhibit a power-law distribution, indicating the presence of opinion leadership.

In the next section, I would introduce Twitch since the platform itself can be unfamiliar to some readers. Then, I describe why should we study Twitch politics. Next, I introduce why political actors choose Twitch as their means to deliver political messages with an emphasis on

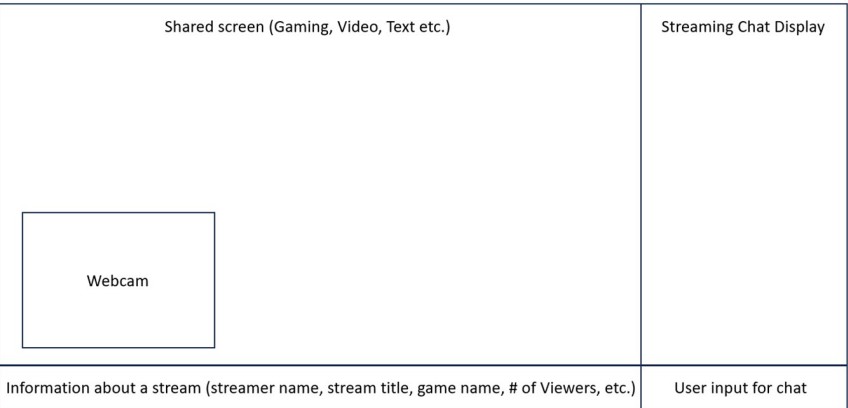

**Fig 1. Schema of Twitch livestreaming screen.**

streaming chat. After that, I introduce three questions to understand Twitch politics and suggest the answers to those questions step by step.

## 2 What is Twitch?

Twitch is about a ten-year-old platform with 140 million unique worldwide monthly visitors and 41.5 million unique US monthly visitors in 2021 [16]. The popularity of the platform looks more impressive when we compare it with traditional media: its monthly viewership in 2018 are comparable to those of some traditional cable TV networks in the US [17].

Fig 1 presents a schematic layout of a typical Twitch stream interface. The shared screen section illustrates how streamers can display gameplay, videos, or other content to engage with their audience. A webcam box in the lower-left corner represents the streamer's live video feed, where they often use a microphone to provide commentary or interact with viewers. To the right of the shared screen is the streaming chat display, where viewers post messages in real time, contributing to the live interaction. At the bottom, the layout includes details about the stream such as the streamer's name, stream title, game being played, and the number of viewers, as well as a text box for users to input chat messages.

Several scholars from multiple disciplines have investigated Twitch from different angles ranging from relatively more macro-level approaches [18, 19] to studies that focus on technology-oriented traits [20, 21] or Twitch affordances [22–24], such as streaming chats [25, 26] and monetization [27]. Macro-level studies on Twitch as a platform economy point out that the flourishing of the platform has been possible because of both demand and supply for streaming [18, 19].

Other scholars examine the platform more in detail. Some focus on the Twitch technologies themselves [21] and its influence on user network [20], while others address the importance of social affordances of Twitch [23, 24] arguing the social structure surrounding those technologies also matters. Sjöblom et al. (2019) thoroughly examines Twitch's affordances while stressing the important role of webcams and audio as they are playing a huge role to establish bonds between streamers and their audiences. Similarly, Jackson (2020) also addresses the power of the synchronous nature that strengthens the perception of intimacy toward streamers, which is the root of their digital persona.

Out of various Twitch affordances, the streaming chat function has received the most scholarly attention [22, 25, 26]. Streaming chat is essential for audience engagement in live

streaming and allows viewers to communicate with the streamer and other participants in the same stream, creating a community that can act as a virtual third place for Twitch users. Affordances related to monetization have been seriously considered by scholars as well [24, 27] as they are the basic building block that buttresses the Twitch platform economy by providing monetary incentives to pursue streaming as a professional career [18, 19].

The political dynamics on Twitch have been studied with a focus on gender politics [28–32] and the role of "emotes" in live streaming political events [33]. Research has highlighted the presence of misogynistic elements within the video game live streaming environment, influenced by policies on sexual content [30] and community guidelines regarding the appearance of female streamers [28, 32]. These factors contribute to the derogation of female gaming streamers [29] and impose additional demands for emotional labor related to affection [28, 31]. Furthermore, the use of "emotes" in the context of political events has been examined, revealing their use as a tool for distracting other viewers [33].

Building upon the existing literature, this study aims to identify political actors on Twitch and analyze their behavior, particularly focusing on the dynamics within streaming chats. While the analysis is centered on a specific timeframe and may not fully capture incidental political content from apolitical streamers, it seeks to provide a deeper understanding of the political landscape on Twitch. By highlighting the significance of these interactions, this study aims to shed light on their broader implications, which will be further explored in the following section.

## 3 Why Twitch?

Why should social scientists care about Twitch, a platform primarily known for its entertainment orientation? While there has been some research on Twitch politics, its political dimensions deserve further exploration. In this section, I will introduce a few episodes that underscore the importance of studying Twitch politics. The first and most famous episode is a series of streams by Alexandria Ocasio-Cortez, also known as AOC. Alexandria Ocasio-Cortez's series of gaming streams on Twitch in October 2020, which aimed to get out the vote for the incoming presidential election, was a huge success, with peak viewership of over 400,000 viewers [34]. A later gaming stream with Canadian MP Jagmeet Singh and other Twitch streamers raised $200,000 for charity [35, 36]. Meanwhile, in her latest stream on Twitch, Alexandria Ocasio-Cortez discussed a political issue—the GameStop stock and Robinhood app issue—with guests including Alexis Goldstein, Alexis Ohanian, and TheStockGuy. They expressed their opinions on the issue and the financial system in general, with AOC advocating for systematic financial reform in the US. The stream was successful, with around 200,000 peak viewers.

Other than politicians, there are other political actors in Twitch: Twitch streamers who stream political content. The best example of a renowned political streamer would be "HasanAbi". His US 2020 presidential election marathon stream on November 3rd, 2020 for 16 hours had 225,000 viewers at its peak [7]. Viewership of his election day Twitch stream was even comparable to the major media outlets when we use "Total Hours Watched" as a criterion to compare what media outlets people have chosen on election day: the stream's portion (4.9%) is almost similar to that of Fox News (6.5%) [37]. The huge success of the election day stream by HasanAbi invited the attention of various news media outlets on the potential of Twitch as a political conduit, along with the huge success of AOC's few streams. As he clearly states in the interview with New York Times, his intention of the Twitch stream has been mainly political—he wanted some platform to congregate people every day and deliver his political opinion [7]. Even entertainment-oriented streamers sometimes speak up about

political issues, with a notable example being the Black Lives Matter protests in mid-2020. Some streamers expressed their opinions during live streams or produced videos related to the issue, while others held fundraising for Black Lives Matter [13]. Twitch politics is not confined to political streamers or viewers, but a broader pool of users and streamers on the platform.

Theoretically, studying Twitch politics grants a novel opportunity to understand how new technologies in the social media environment affect the patterns of political communication. Twitch, an entertainment-focused live streaming platform, offers an environment for us to study how emerging technologies such as "live-streaming" and "streaming-chat" influence the dynamics of political communication.

What is missing in this study is the analysis of the large population who might encounter political content incidentally, not by watching explicitly political streams. "Incidental exposure" to political news or content is well-known to play a role in agenda setting on social media [38], contributing to an individual's political knowledge acquisition, albeit with some limitations compared to intentional exposure [39–41]. However, incorporating this dimension into the current study posed significant methodological challenges. Observing incidental exposure would require analyzing chat posts across a vast number of streams to identify political content, a task made highly challenging by the scale and variability of Twitch data. This limitation highlights the difficulty of capturing incidental political communication where apolitical streamers may subtly or sporadically engage in political discourse. While this omission is acknowledged, the study's focus remains on overtly political actors, where data collection and analysis are more feasible.

## 4 What makes political content creators choose it?: The importance of streaming chat function and systematic dependence on audiences

People who want to create political content on a streaming platform may choose Twitch because it provides an easy environment to start streaming. Creating video content for You-Tube can be efficient compared to producing text content [9], and Twitch live streaming might be more efficient as it does not require post-shooting edit as most of YouTube videos do, although live-streaming entails some amount of time to prepare the show. The user-friendly interface of Twitch and the availability of the app "Twitch Studio" make it easy for anyone to start streaming using their PC or mobile phone. The only requirements are an electronic device to run the program and objects to show viewers, primarily the streamer themselves.

As the previous literature thoroughly examines, the existence of various ways to pursue financial revenue also makes Twitch more attractive [24, 27]. Twitch offers direct revenue streams such as subscriptions, viewer donations through Twitch currency called Bits, and advertisement revenue. Streamers can also earn money through external sources such as sponsored product placement and third-party donation platforms. Although only handful amount of streamers can earn a stable income, the potential for financial rewards is a factor that attracts users to the platform. Hence, Twitch provides an ideal platform for political content creators to brand themselves as micro-celebrities. By using webcams and microphones [24] to create a personal connection with their audience, they can establish credibility [8] and foster a sense of community through parasocial relationships [42]. This connection can be further strengthened by including links to other social media platforms in their profiles [24], enhancing their overall engagement with the audience. The chat function on Twitch enables the communication between streamers and viewers, creating a space for mass communication that was not possible in traditional media outlets [43]. Viewers can instantly react to what the streamer says or does

by using Twitch emotes or normal languages, which helps create a relatable and accountable relationship between the streamer and the audience [11].

Twitch's real-time chat function provides an attractive feedback loop for political content creators and audiences to have a discussion on political topics in real-time, which can be very attractive to political content creators. While platforms like Twitter and Facebook allow for discussion on political topics, there is usually a time lag between users' responses. However, Twitch's chat function allows political content creators to express their opinions in real-time while engaging with their audience, giving the impression of a live discussion. This engagement can attract potential viewers interested in expressing their opinions on contentious political topics. Overall, the effective streaming chat function which enables political streamers to have a real-time discussion with their audiences can be the most important feature of Twitch politics. Exploring the role of streaming chat on political communication in the social media context can have some implications for the political communication literature as we little know how it affects inter-user communication patterns.

The other important trait of Twitch is that the creation of the content is also dependent on viewers at the same time. Not to mention the audience is a source of revenue that streamers are making [27], their content creation itself is also inherently dependent on viewers and their chat posts because of the real-time nature of live streaming [12]. Viewers' reactions during live streaming are crucial for political content creators, as their content tends to be talk show-oriented and requires smooth communication with viewers through streaming chat. Without active participation from viewers, political streaming cannot be successful and will only be a manifestation of the creator's political opinion. The audience users of Twitch have powerful agency and their participation is essential for the success of streamers. This highlights the need to study the audiences of political streams and their interactions with each other, not just the political streamers.

## 5 Three questions: Who are they, what types of political content are discussed, and how do they interact?

In this paper, I answer three questions on Twitch politics. First, who are the political streamers on the platform? To study Twitch politics, it is imperative to identify political streamers first from the population of streamers, who are mostly non-political. I have retrieved extensive lists of streamers who may stream political content and their information using Twitch API. Based on the information, I have leveraged supervised machine-learning techniques to find political streamers from the retrieved lists.

Second, what types of political content are discussed during streams? Understanding the nature of political discourse on Twitch is crucial, as it provides insight into what political content exists on the platform and how it is addressed by streamers. Do streamers cover topics widely discussed in traditional media, or do they highlight issues that are underrepresented elsewhere? Analyzing their topic choices can reveal the unique patterns of political communication on Twitch. To explore these questions, I conducted a series of text analyses on a dataset of streaming chat posts collected during a specific timeframe. This timeframe captures a snapshot of Twitch politics, focusing on the political topics most prominent during that period. The dataset contains 33,649,628 chat posts collected from 478 political streamers, out of a total of 574 identified political streamers. Data collection was conducted in compliance with Twitch's API terms and conditions. All user-identifiable information has been anonymized to protect user privacy, and no identifiable data will be shared in this analysis or in any resulting publications.

Third, how do political actors, both political streamers and their audiences, interact with one another? In order to understand the political communication behaviors of Twitch users, which includes both streamers and their audiences, it is necessary to accurately capture the communication networks. I would focus on inter-audience communication in a stream that is done by mentioning others' usernames. How do Twitch users in political chat rooms behave in terms of communicating with each other via streaming chat? By constructing and analyzing reference networks of each political stream, I will study political communication networks in Twitch political streams.

## 6 Question 1: Who are the political streamers?

In this section, I find political streamers out of extensive lists of streamers that are retrieved via Twitch API using the supervised machine learning method. In the first subsection, I briefly introduce Twitch API and how I have utilized it to get extensive lists of streamers. Then, I introduce the coding scheme to classify political streamers and the classified list of political streamers.

### 6.1 Using Twitch API to retrieve streamer lists

I collected streaming records from Twitch during the period of 21-08-31 00:25 EST to 21-09-08 16:44 EST, focusing on broadcasts tagged with the game categories "Just Chatting", "Talk Shows and Podcasts", and "Politics", using Twitch API. These categories were selected because they are likely to contain streams with political content through the process elaborated in Section A of S1 Appendix. While most streamers do not broadcast continuously, collecting data over a week provides a reasonable timeframe to capture streamers who are serious about their content and are likely to stream at least once in that period. This one-week window was chosen to identify such political actors, acknowledging it as a practical starting point for capturing a snapshot of political content on Twitch. The streaming records include information such as user names, user IDs, and stream titles. In total, I identified 53,550 unique user ids for "Just Chatting", 5,853 for "Talk Shows and Podcasts" and 324 for "Politics". Additionally, I gathered streamers' profile descriptions, which offer insights into the themes they regularly address. By merging the stream titles and profile information, I constructed a dataset representing the content and focus areas of these streamers.

### 6.2 Finding political streamers using supervised machine learning techniques

I have used a supervised machine-learning technique to identify political streamers from the user data I have collected. The political streamers are defined by the following coding rules refer:

1. If a streamer has broadcasted political content at least once (i.e. "Texas Abortion Law is a shame"), I coded her as a political streamer.

2. If a streamer profile shows that she broadcasts about or is at least interested in politics (i.e. "I sometimes talk about politics"), I coded her as a political streamer.

3. If a streamer explicitly identifies their political interest or partisanship on her profile (i.e. "I am Leftist", "This is conservative podcast"), I coded her as a political streamer.

4. If a broadcasting title or streamer profile contains a representative term or hashtags of specific political movements (i.e. "#BLM", "#FreePalestine"), I coded her as a political streamer.

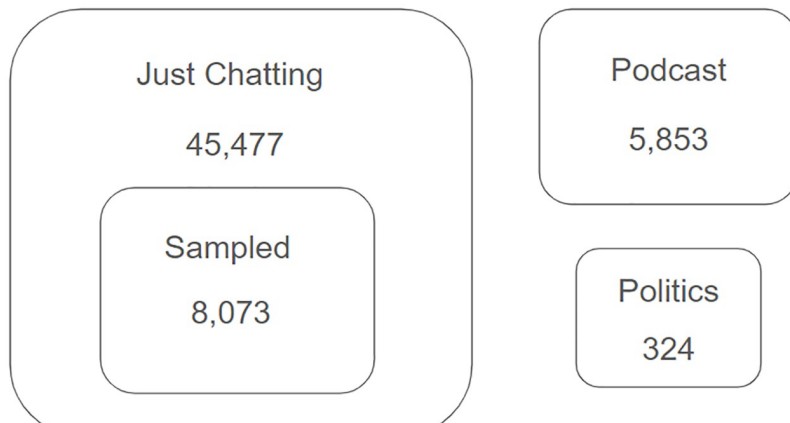

**Fig 2. Venn diagram of Twitch streamer data.** The figure illustrates the distribution of identified Twitch streamers across three key categories: "Just Chatting," "Talk Shows and Podcasts," and "Politics." Out of 45,477 streamers categorized under "Just Chatting," a subset of 8,073 was sampled using a keyword filtering approach based on terms commonly associated with political content. The smaller circles represent the total number of streamers in the "Talk Shows and Podcasts" (5,853) and "Politics" (324) categories.

As I already know that there would not be as many political streamers in the list of "Just Chatting" compared to "Talk Shows and Podcasts" and "Politics" as described in Section A of S1 Appendix, I adopt the following strategy to produce training sets with enough target values, political streamers, to ensure the model gets trained properly. First, I have hand-coded all components from the "Talk Shows and Podcasts" and "Politics" sets in Fig 2. Then, I extracted the top 30 words from the combined text data of political streamers found in the first step and filtered out nonessential words, such as articles and be-verbs, and other common words that are widely believed to be frequently used by most Twitch streamers (i.e. stream, live, etc.). I have used these keywords as a filter to sort out observations that are more likely to be coded as political from "Just Chatting" data. The process gives the "Sampled" set with 8,037 streamers that is right inside of the "Just Chatting" set in Fig 2. I hand-coded all of these streamers and identified 225 political accounts. The whole process gives me a total of 14,250 hand-coded train data with 550 political streamers. By using the labeled data, I fitted the supervised machine learning classifier and was able to identify 574 political streamers in total. Details about the machine learning model training process are described in Section C of S1 Appendix.

### 6.3 Descriptive features of political streamers

Based on the data retrieved from the Twitch API, we can outline key descriptive features of political streamers. Fig 3 illustrates two histograms: the distribution of the dates when streamers joined the platform (top) and the distribution of the number of views on their profiles (bottom). As expected, the majority of political streamers joined the platform relatively recently, with a notable concentration on the right side of the x-axis. Interestingly, a small number of streamers joined quite early, some joined as early as 2008. Since Twitch officially launched in 2011, this suggests that these users were originally members of Justin.tv, Twitch's predecessor. Thus, while most political streamers have a relatively short broadcasting history, a subset of streamers has been on the platform for a significantly longer period, potentially indicating a more established presence.

The second histogram displays the log-transformed distribution of profile view counts, serving as an indicator of streamer popularity. The distribution is highly skewed, with a small

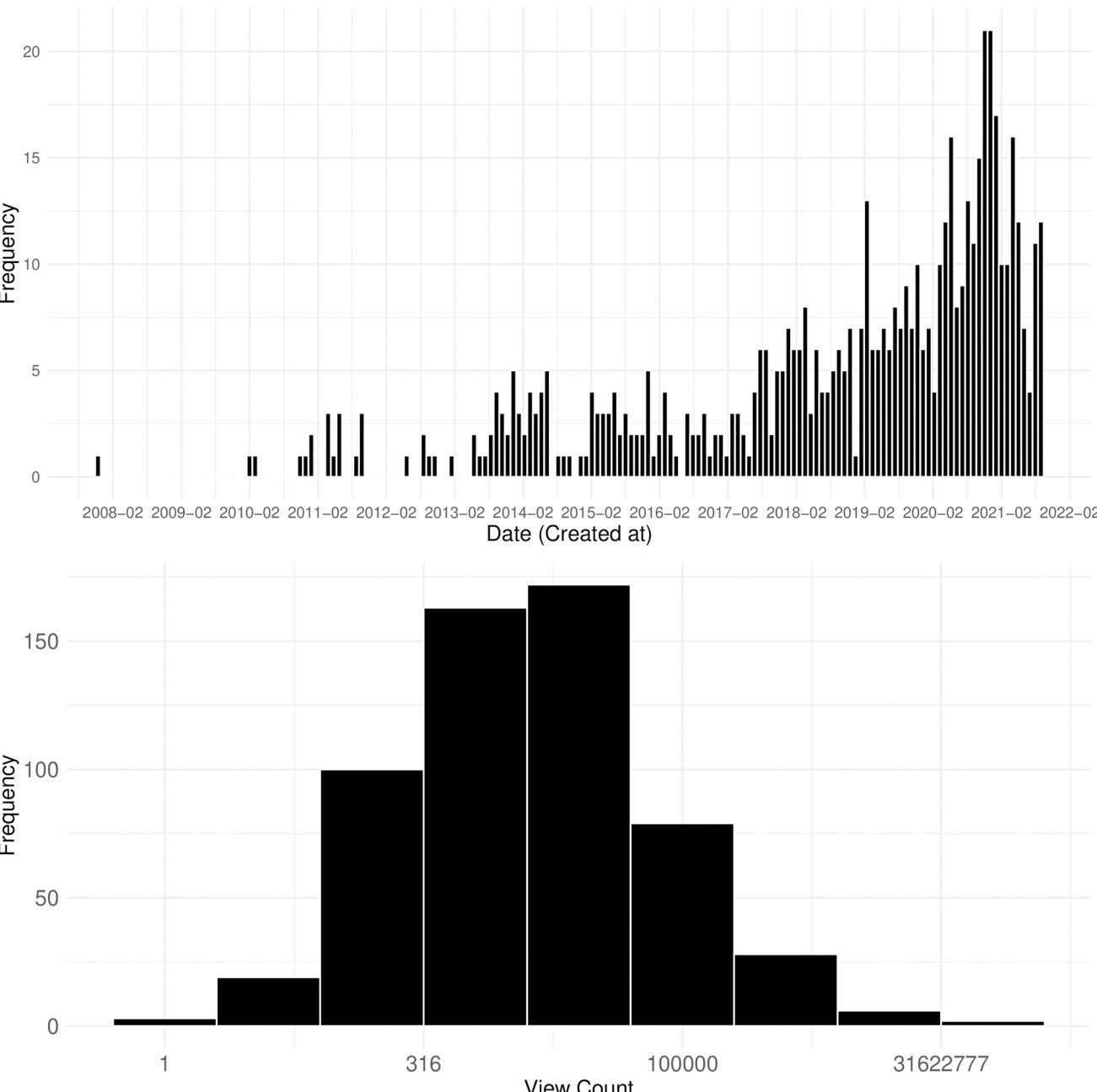

**Fig 3. Histograms of account creation date (above) and profile view counts of political streamers.** The below histogram's x axis is log-transformed while labels show the raw values.

number of streamers amassing substantial view counts, while the majority remain relatively obscure. The median profile view count is 3,228, and those in the lower 150 range have fewer than 316 views, indicating that their popularity on the platform is minimal. A D'Agostino's K-squared test confirms the skewness of the distribution (skew = 16.2, z = 30.146, p < 2.2e-16), highlighting the significant disparity in the visibility and popularity of political streamers on Twitch, with only a small fraction enjoying widespread attention.

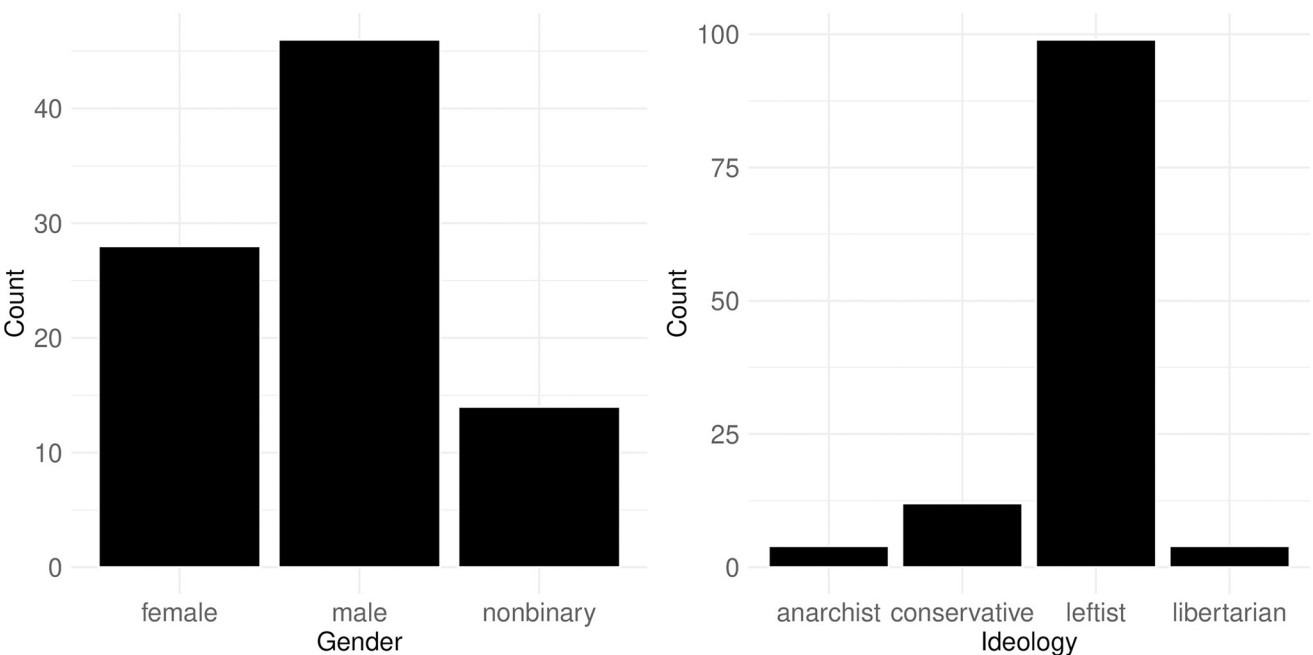

**Fig 4. Barplots of gender (left) and ideology (right) of political streamers.** As most of the streamers do not reveal those traits in their profile, the number of values in the historgram is very limited (N = 88 for gender, N = 119 for ideology).

Fig 4 presents the personal traits of political streamers as reported in their account profiles, specifically focusing on gender (left) and ideology (right). It is important to note that a significant number of streamers do not disclose these traits in their profiles, resulting in limited sample sizes for the histograms (N = 88 for gender and N = 119 for ideology), which restricts the generalizability of the findings. The coding rules for the values represented in these figures are detailed in Section B of S1 Appendix.

The analysis reveals a male dominance among political streamers; however, a notable observation is that approximately 10% of those who disclosed their gender identify as nonbinary. Regarding ideological alignment, the majority of streamers who reported their ideology classify themselves as leftist, with very few identifying as anarchist or libertarian. This highlights the predominance of leftist views within the sample of political streamers on the platform.

## 7 Question 2: What types of political content are discussed during streams?

### 7.1 Collecting chat posts from political streamers' live streaming using 'chat bots'—Descriptive figures

As a first step to studying political discussion of identified political streamers, I have used chat-bots that get connected to each stream and exist in the channel until it gets offline [22] to collect chat posts from streams provided by political streamers I have identified. The bots allowed me to download all chat posts in the stream's associated IRC (Internet Relay Chat) with various information, ranging from the text of chat posts to the user-name of the sender [22]. Using the pipeline, I have collected chat posts of political streamers from 2021-12-11 to 2022-03-25.

Due to the time lag between the identification process of political streamers and actual chat post data collection, there has been some loss in the number of political streamers. I was able

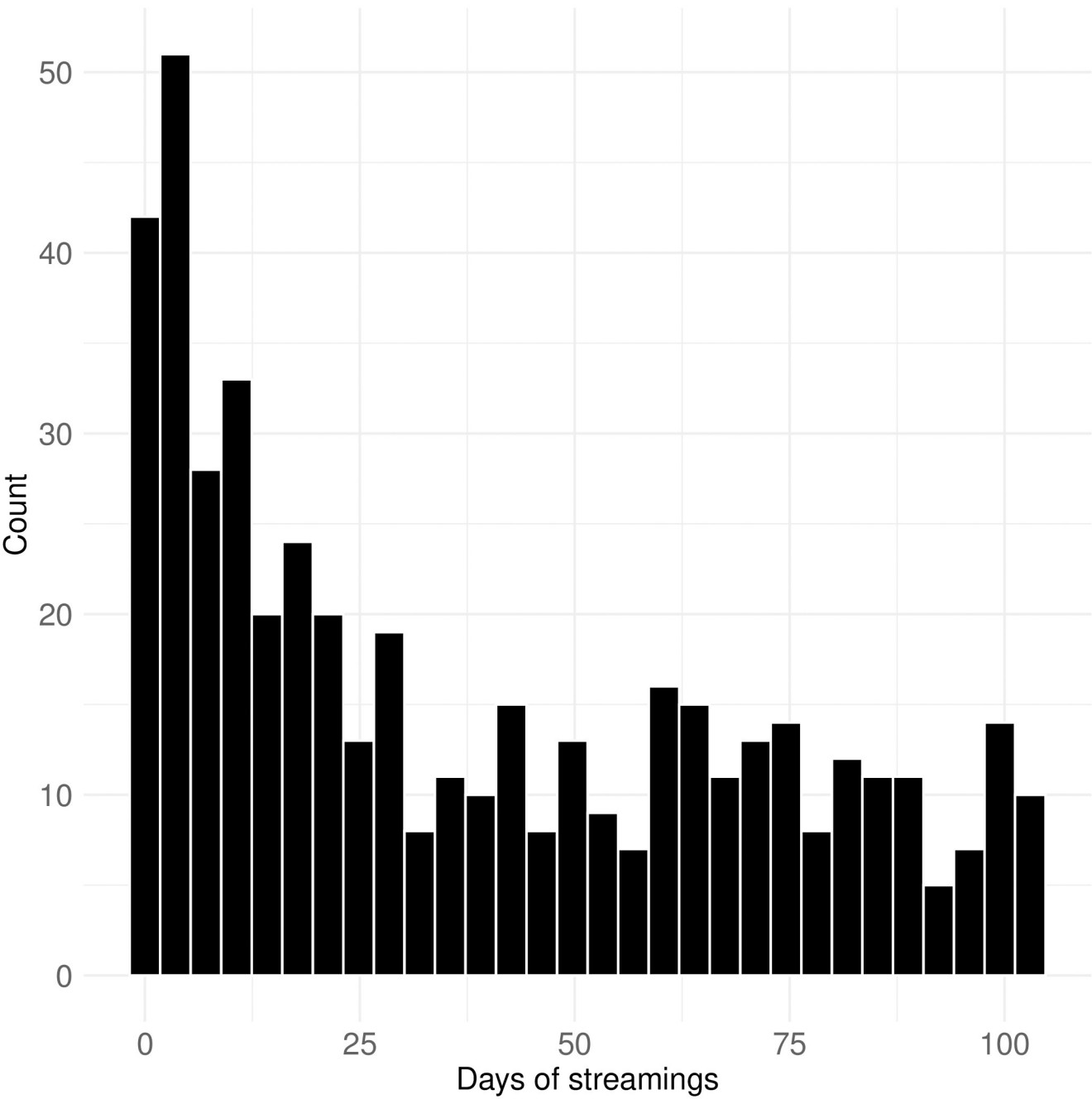

**Fig 5. Frequency of streamings by political streamers.** This figure shows how often political streamers started streams during the data collection period (December 11, 2021, to March 25, 2022). While the data spans 105 days, a notable number of streamers were highly active, with 342 streaming on more than 10 days and 228 streaming on more than 30 days.

to collect chat posts of 478 political streamers out of a total of 574, which means the survival rate is above 83%. The reasons for the loss can be diverse. Some streamers might have quit streaming at all, or some might just want to cease it for a while.

Fig 5 illustrates the daily frequency of political streamers starting their streams during the data collection period, which spanned from December 11, 2021, to March 25, 2022. Since this

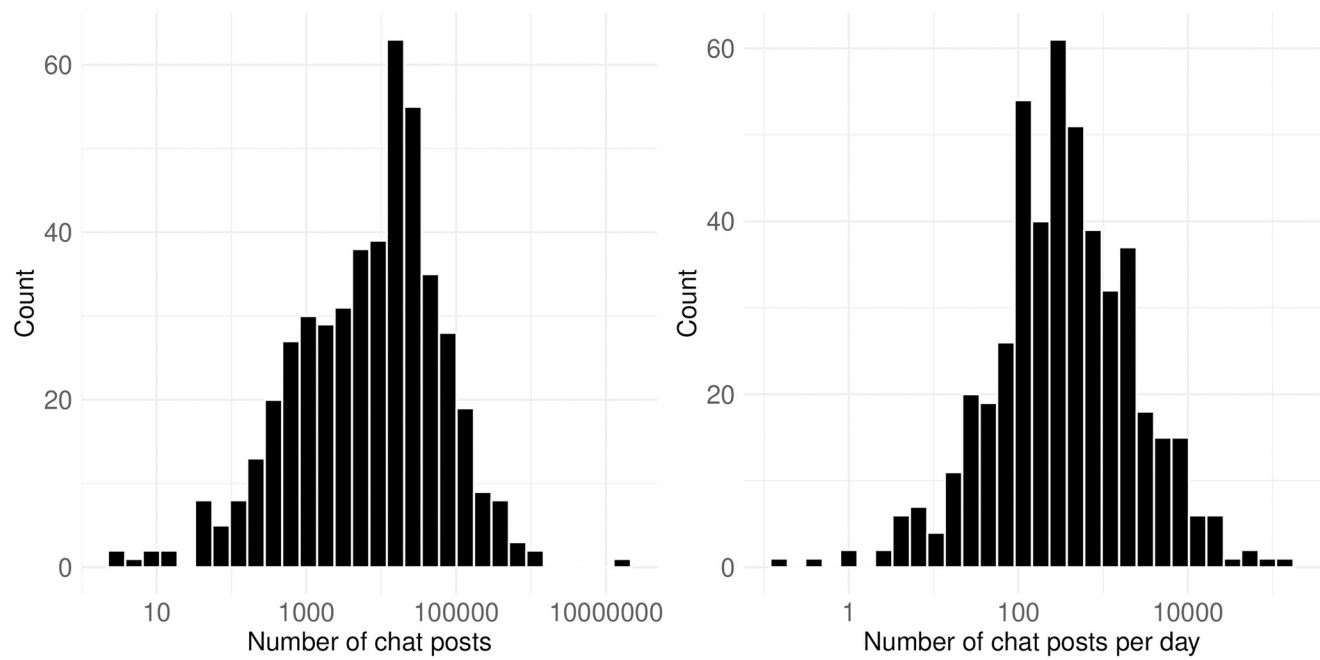

**Fig 6. Number of chat posts.** This figure shows the distribution of chat posts per streamer's channel, both in total and on a daily basis. The x-axis is log-transformed while labels show the raw values, approximating a normal distribution with extreme values on both ends.

period covers 105 days, the maximum value in the plot is naturally 105. While some streamers were very active throughout the period, others were less so. D'Agostino's skewness test (skew = 0.519, p-value < 0.0001) indicates a slight positive skew in the distribution of streamer activity, reflecting the fact that most streamers were moderately active, but a few streamed more frequently. Specifically, 342 political streamers streamed on more than 10 days, and 228 of them streamed on more than 30 days over the three-month period. This suggests that most of the data captures regular interactions between viewers and streamers, although some streamers broadcasted infrequently.

Fig 6 shows the number of chat posts on each streamer's channel, both in total and on a daily basis. The x-axis log transformed histograms approximate a normal distribution, with a small number of extreme values on both ends. The total number of chat posts collected from all channels is 33,649,628. The channel with the most chat posts has 16,945,559, while another channel has only 2 posts from a single viewer. The mean and median values are 70,396 and 2,430, respectively. On a daily basis, where the number of chat posts per channel is divided by the number of streaming days, the mean and median values are 2,224 and 339, respectively. This suggests that, for a political channel in the middle of the distribution, we can expect at least 350 chat posts per stream. This indicates that most political channels receive some level of audience interaction, reflecting feedback loops in the political streams.

Fig 7 illustrates the distribution of unique usernames that have posted at least one chat post on a channel. The histogram indicates that the distribution is heavily skewed to the right, as confirmed by D'Agostino's skewness test (skew = 19.515, p-value < 0.0001), showing a significant departure from symmetry. The most popular channel boasts 267,206 unique users, while the least popular has only one unique user. In total, there are 646,073 unique users across all channels, with mean and median values of 1,351 and 70, respectively. The higher mean compared to the median is largely influenced by extreme values on the right-hand side of the

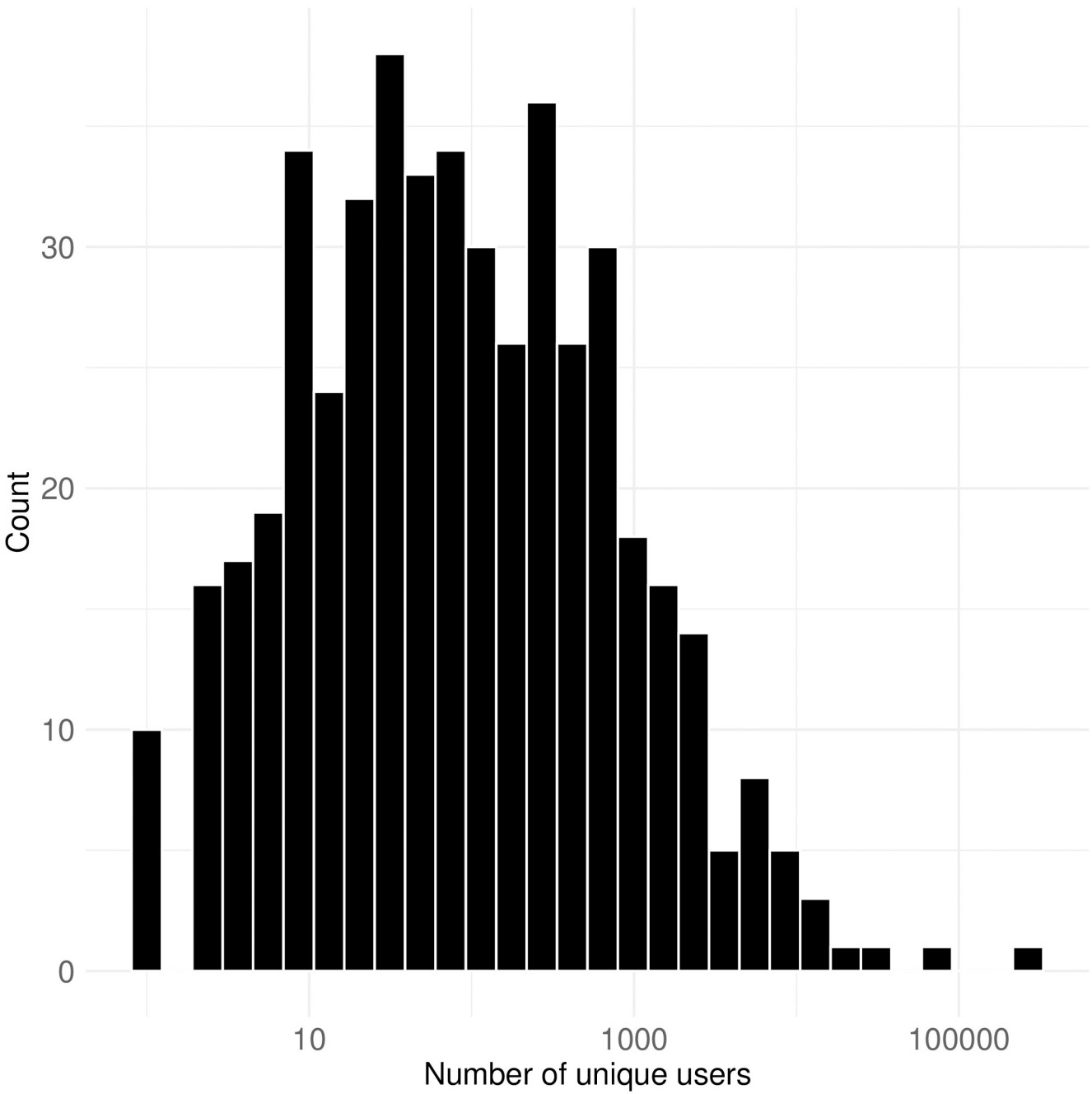

**Fig 7. Number of unique users in a channel.** This figure shows the distribution of unique users posting at least one chat message per channel, with the x-axis is log-transformed while labels show the raw values. The most popular channel had 267,206 unique users, while the least had only one.

distribution. Therefore, most of channels are likely to have around 70 unique users who have posted at least one chat post during its streaming. While a smaller audience size can foster active political discussions among users and streamers, establishing parasocial relationships [42] and gaining micro-celebrity status [8] may be challenging due to their dependence on popularity.

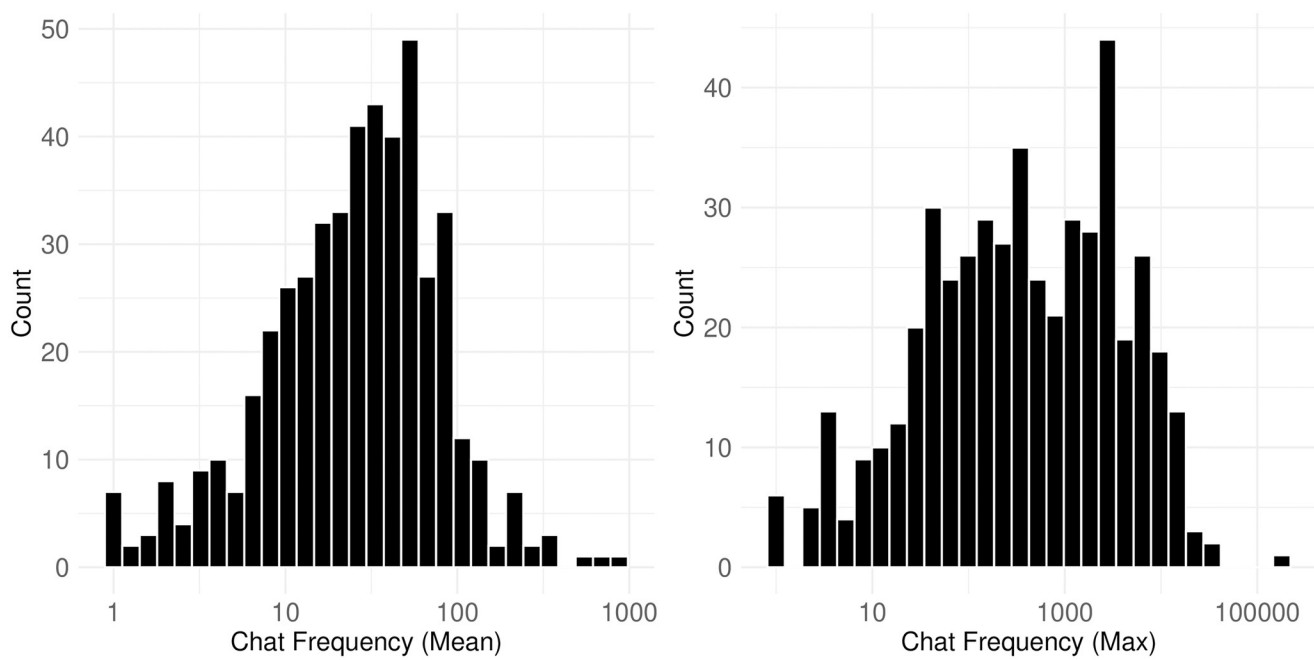

**Fig 8. Frequency of chat posts.** This figure displays user-level chat frequency, with the x-axis is log-transformed while labels show the raw values. The median values are 28.5 for mean chat frequency and 377 for maximum.

Fig 8 shows the frequency of chat posts at a user level, with the x-axis is log-transformed while labels show the raw values. The mean chat frequency of all users in a channel displays a nearly normal distribution, whereas the maximum chat frequency has a weak left skew, as confirmed by D'Agostino's skewness test (skew = -0.307, p-value = 0.0065). The mean and median values for the mean chat frequency are 45.3 and 28.5, respectively, while those for the maximum chat frequency, which shows the most active user from each channel, are 2,475 and 377. These patterns suggest that most users are participating in the chat repeatedly. However, the significantly high number of chat posts from a single user suggests the presence of bots, which are commonly used in Twitch streams.

## 7.2 What is discussed in political Twitch streams?: Topic models of the chat post data

In this section, I will discuss the content of political streams by conducting topic modeling on chat post data. Topic modeling, a tool widely used in both social science and computer science [44, 45], helps us observe general trends in the text corpus by examining broad categories among chat posts. Through topic modeling, we can identify topics, which are sets of words that frequently occur together within documents [45].

To manage the large volume of chat posts, I applied a 5% sampling rate for each political stream. This rate was chosen as a pragmatic solution to the computational limitations of processing over 33 million chat posts locally. Analyzing 100% of the data exceeded the available computing resources, so I conducted preliminary test runs with different sampling rates. After testing 10%, 8%, 7%, and 5%, the 5% rate was selected as it offered the best balance between computational feasibility and preserving the diversity of content for meaningful analysis. The number of chat posts collected varies significantly by stream due to differences in audience size and broadcasting frequency, so I sampled 5% from each channel rather than at an

**Table 1. Topic proportions.** Proportion for political chat posts and all chat posts are calculated by dividing the number of topics with the number of political topics, 45, and the total number of topics, 150.

| Category | Number of topics | Proportion (Political) | Proportion (Total) |
|---|---|---|---|
| International issues | 10 | 22% | 6% |
| US politics | 10 | 22% | 6% |
| Identity politics | 9 | 21% | 6% |
| Ideological debate | 7 | 16% | 4% |
| Politics in general | 4 | 9% | 3% |
| Public health and politics | 2 | 4% | 1% |
| Environmental issues | 2 | 4% | 1% |

aggregate level. This rate provided a manageable dataset for text processing and topic modeling while preserving content diversity. In total, 1,682,739 chat posts were sampled. I identified a topic as political if one or more politically relevant keywords were among the top 15 keywords within a topic (See Section D of S1 Appendix for more details). Based on this criterion, I found 44 political topics out of a total of 150 topics. This means that 30% of the topics identified through the analysis are political. Comprehensive lists of political topics are provided in Section E of S1 Appendix and all topic lists are provided in Section F of S1 Appendix.

I have categorized political topics into seven categories based on the political keywords in those topics: 1) International issues, 2) US politics, 3) Identity politics, 4) Ideological debate, 5) Politics in general, 6) Public health and politics, and 7) Environmental issues. The coding rule for topic categorization is provided in detail in Section D of S1 Appendix. Table 1 shows the number of topics in each category and the proportion each category represents in the corpus of political chat posts and all chat posts.

One of the most frequently appearing political topics in Twitch streams are related to international issues, which constituted 22% of the political chat posts and 6% of all chat posts. Fig 9 provides more details about these international topics. The prevalence of international issues, particularly those concerning the Russian invasion of Ukraine, is reflective of the specific time frame during which data was collected. Six topics directly reference the Ukraine war or Russia, and other topics address international disputes involving NATO-related countries (e.g., "International disputes") and the China-Taiwan conflict ("China and Taiwan"). These findings offer valuable insights into the political discourse on Twitch during a significant global event. However, the analysis is shaped by the timing of the data collection, and additional political topics may emerge across a broader or different timeframe.

As Fig 10 shows, topics related to US politics also appear frequently: the number of topics is 10 and they account for 22% of political chat posts and 6% of all chat posts. As Twitch is the most popular in the United States, this finding is not that surprising. There is a topic about Trump ("Trump and Gender") and Biden ("Biden politics"), the two most popular politicians in the United States. Alexandria Ocasio Cortez was also mentioned in chat posts ("AOC"). Topics related to controversial violent events, such as Kyle Rittenhouse incident ("Kyle Rittenhouse") and police violence ("Florida Police Violence"), and the Black Lives Matter protest ("BLM") appear in political streams as well.

Interestingly, topics related to identity politics appear almost as frequently as those related to US politics and international issues. There are 9 topics, accounting for 21% of political chat posts and 6% of all chat posts. Fig 11 illustrates the identity politics-related topics. The most frequently appearing topic in the political chat post corpus is related to trans rights ("Trans Right"). Gender issues are also widely covered, with three topics related to gender politics.

## Ranking of topics on international issues

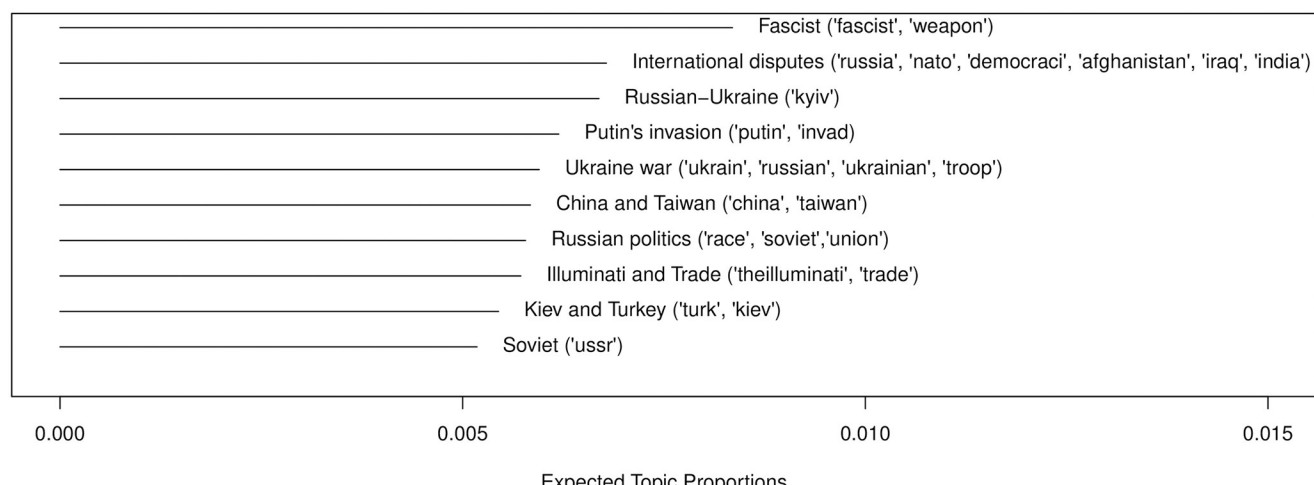

**Fig 9. Topic ranking: International issues.**

Racism is another significant issue in political streams, with three topics addressing racism or race-related issues ("Asian issue"). This finding aligns with the literature, which suggests that Twitch and other online spaces with streaming chat functions can act as virtual third places, revealing users' identities [26], and fostering a sense of community through parasocial relationships between streamers and audiences [42]. Audiences may feel that Twitch political streams provide a space where they can reveal their identities and comfortably discuss related issues.

Topics related to ideological standings are also present. There are 7 topics, comprising 16% of political chat posts and 4% of all chat posts. Fig 12 provides more details about these topics of ideological debate. More than half of the topics reference leftist ideologies, while only two

## Ranking of topics on the US politics

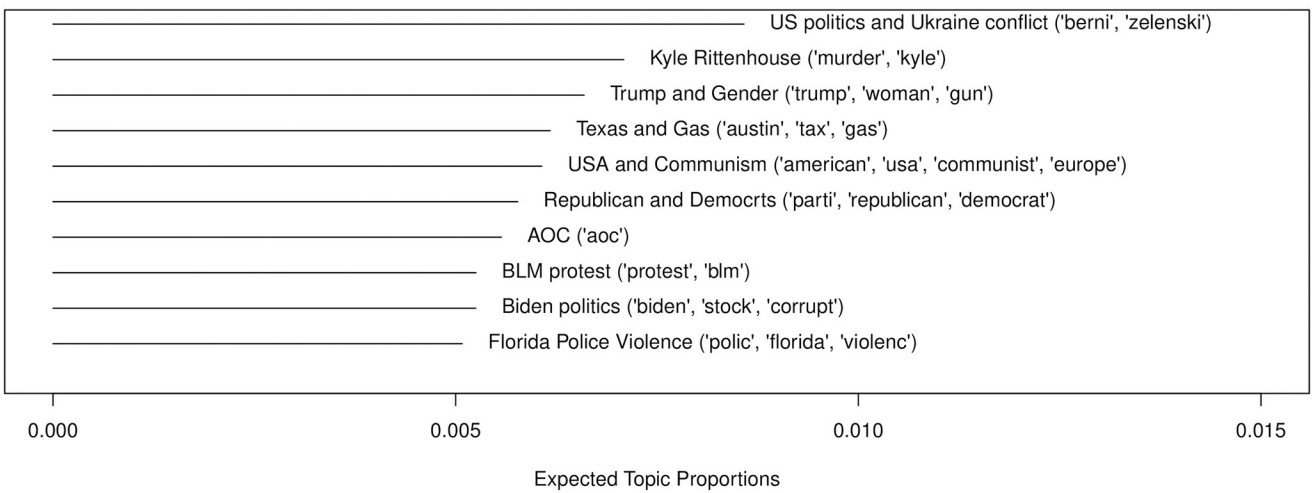

**Fig 10. Topic ranking: The US politics.**

**Ranking of topics on identity politics**

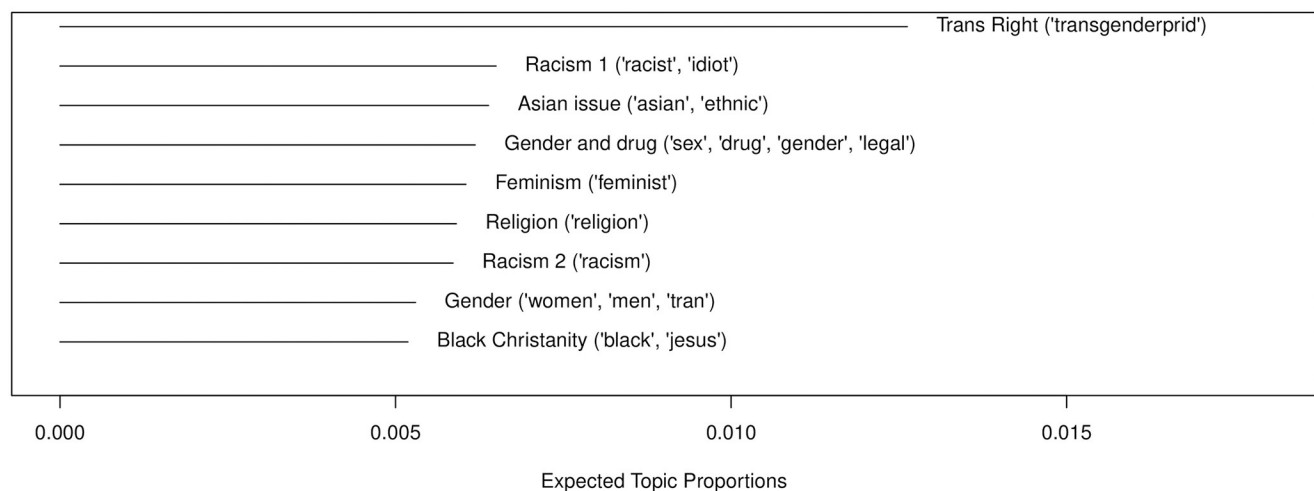

**Fig 11. Topic ranking: Identity politics.**

topics indirectly relate to conservative or right-wing ideas. This may suggest that right-wing-oriented political streams engage in self-reinforcing communication by condemning leftist political enemies. Additionally, the categories "Public health and politics" and "Environmental issues" each have two topics, accounting for 4% of political chat posts and 1% of all chat posts, respectively.

One notable aspect is the use of Twitch-specific communication modes, particularly "emotes," in political discussions. Twitch emotes are akin to emojis used in streaming chats. Fig 13 demonstrates that three emotes—Potchamp, Bboomer, and Kkapitalist—are frequently employed in chat posts categorized as political. The images of these commonly used emotes, designed to satirize affluent capitalists (Kkapitalist), Baby Boomers (Bboomer), and to express

**Ranking of topics on ideological debate**

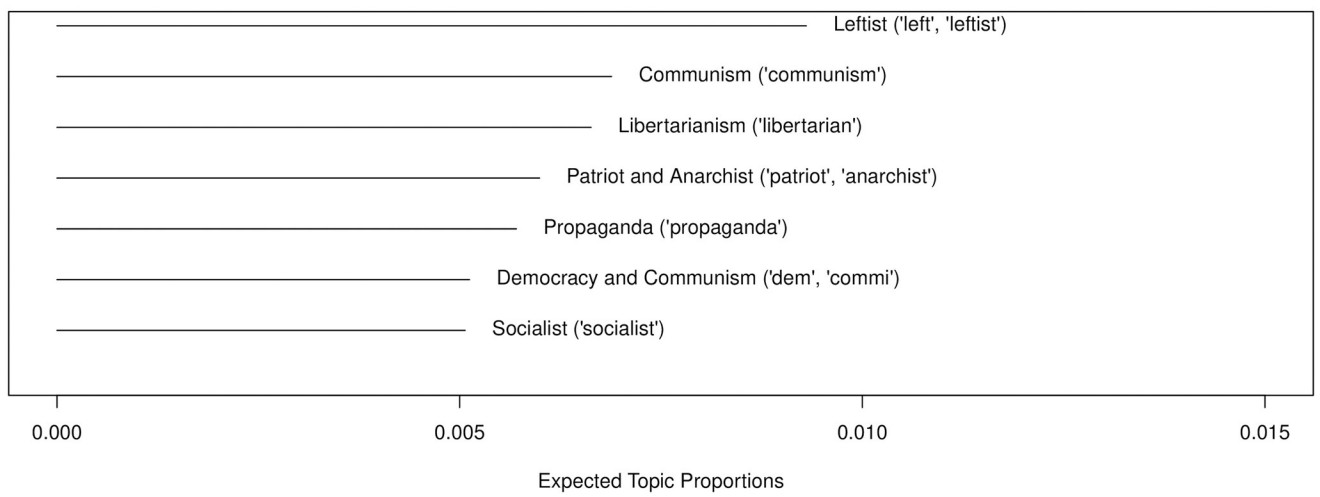

**Fig 12. Topic ranking: Ideological debate.**

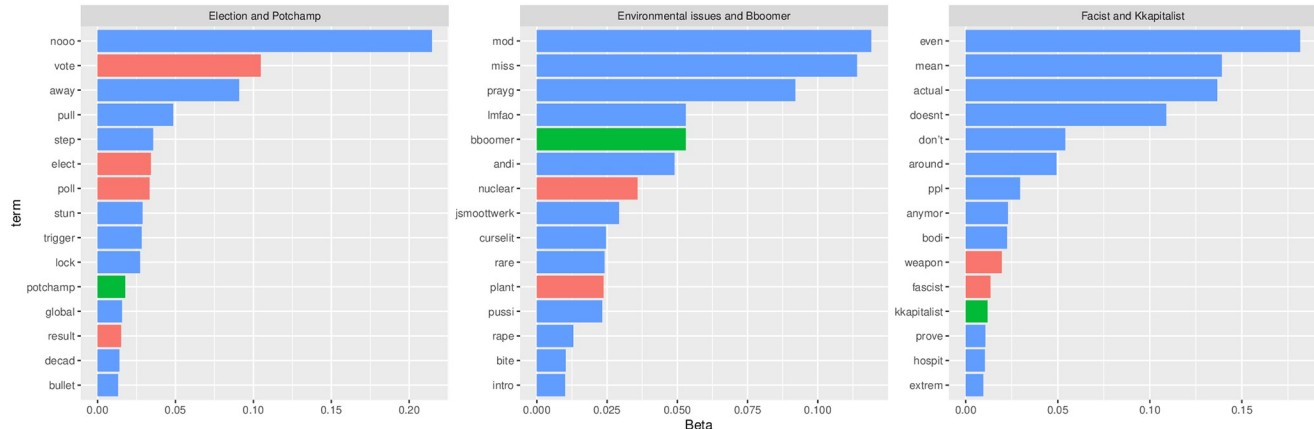

**Fig 13. Emote usages in political streams.** This figure illustrates the use of Twitch emotes in political discussions, highlighting three frequently used emotes: Potchamp (expressing joy), Bboomer (satirizing Baby Boomers), and Kkapitalist (satirizing affluent capitalists). The prevalence of these emotes may reflect certain political inclinations, including anti-capitalist and anti-Boomer sentiments.

joy (Potchamp), can be accessed via the following links: https://www.frankerfacez.com/emoticon/340801-KKapitalist, https://betterttv.com/emotes/5c447084f13c030e98f74f58, and https://www.frankerfacez.com/emoticon/426503-PotChamp. The prevalence of these emotes suggests certain political inclinations among Twitch users, potentially indicating anti-capitalist and anti-Boomer sentiments. Alternatively, this usage can be interpreted with more context. For instance, some users may associate Boomers with environmental concerns, as keywords related to environmental issues often co-occur with "Bboomer" in the topic Environmental issue and Bboomer in Fig 13. Overall, political discourse on Twitch utilizes not only text but also context-specific emotes, which merits further exploration by scholars of political communication.

# 8 Question 3: How do they interact?

## 8.1 Reference network in chat posts within a political stream

Using the collected chat post data, I constructed 'reference' networks to observe how users interact within political streams. In these networks, each user who posted at least one chat message is represented as a node, and directed edges between nodes indicate interactions where one user directly mentions another.

The edges were created based on user mentions in chat messages, which typically use the @ symbol to refer to another user. Specifically, the steps were: 1) I identified all chat posts containing the @ symbol, a convention used to tag or refer to other users. 2) From each such post, I extracted the username that appeared immediately after the @ symbol as the referenced user. 3) A directed edge was then created from the user who posted the message to the referenced user, provided the name after @ matched an existing user in the same chat. This process formed a network where interactions are represented by edges pointing from the referring user to the mentioned user.

Out of 478 active political streamers, I constructed 279 directed reference networks, corresponding to channels where at least one user mentioned another. For the remaining channels, no networks were formed due to the absence of user mentions, as edges cannot be established without references between users.

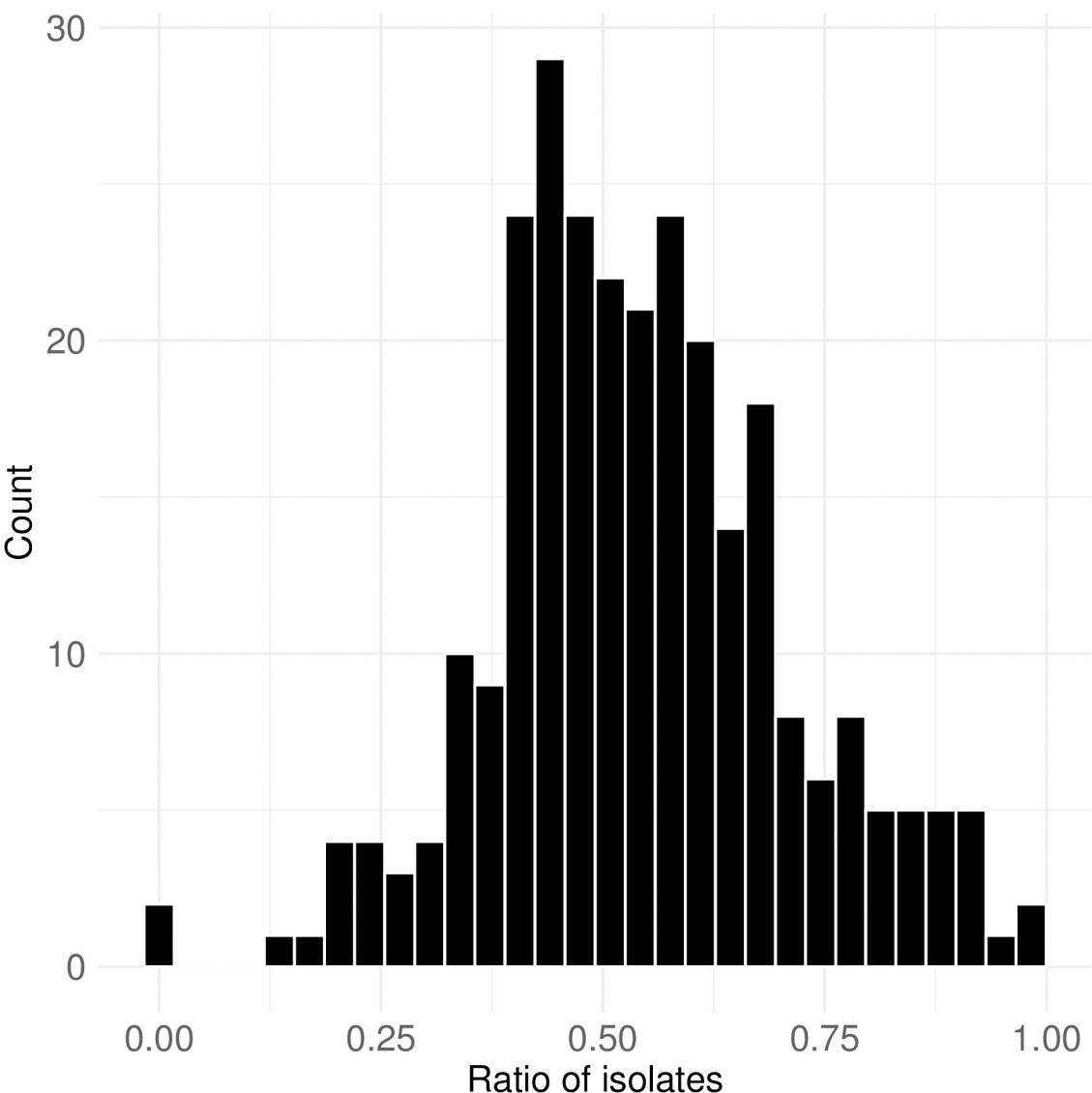

**Fig 14. Ratio of isolates.** This figure shows the distribution of isolate ratios across 279 mention networks, with a mean slightly above 0.5, indicating that nearly half of the users never post a chat with a mention (@).

**8.1.1 Network descriptive statistics.** One notable observation is that reference networks typically include a significant number of isolates. Fig 14 illustrates the distribution of isolate ratios across 279 mention networks. The distribution follows a nearly normal pattern with a mean slightly above 0.5, indicating that nearly half of the users in these networks never post a chat with a mention ('@'). While Twitch is generally oriented toward audience-to-streamer interactions, the lower-than-expected isolate ratio might be due to the unique nature of political streams, which often encourage more peer-to-peer discussion. Many users may engage in conversations with others in the chat rather than solely commenting on the stream content. Notably, in about 40 political streams, over 60% of interactions involve references to other viewers or streamers, suggesting highly engaged, conversational dynamics. This level of engagement is more commonly observed in smaller, more intimate Twitch communities [43],

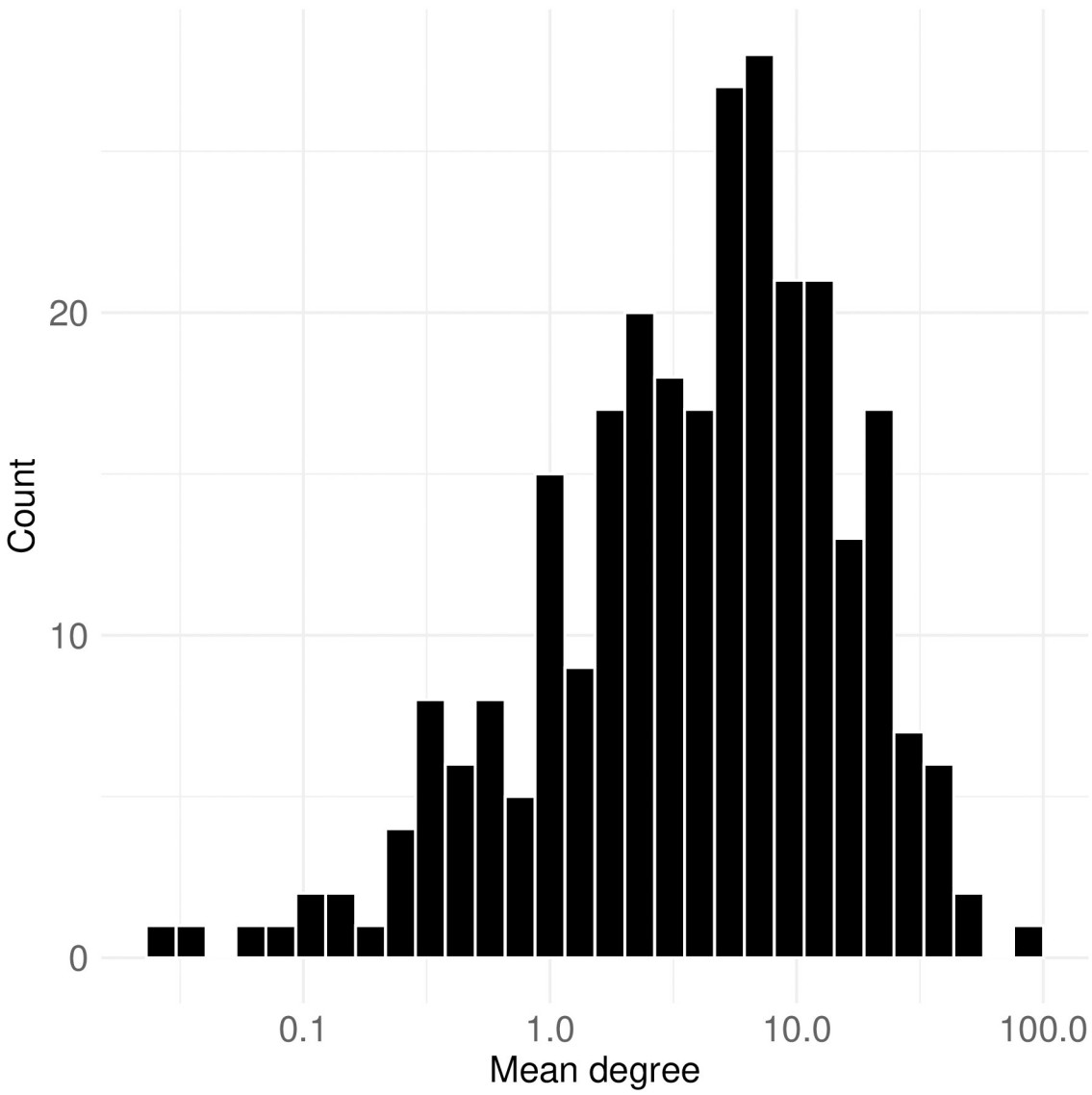

**Fig 15. Distribution of mean node degrees.** The x-axis is log transformed while labels show the raw values. The distribution approximates a normal pattern, with a mean of 7.9 and a maximum mean degree of 84.3, indicating the frequency of chat posts that mention other users.

where users form direct connections with each other. This also aligns with the earlier finding that most channels likely involve around 70 users who have posted at least one chat post (see Fig 7).

Then, how many chat posts mentioning other users do Twitch users send and receive? The histograms depicting the mean degree of all nodes in each mention network are presented in Fig 15. The distribution of logged values indicates a nearly normal pattern, with a mean of 7.9 and the highest mean degree reaching 84.3. Fig 16 further illustrates distributions of maximum node degree, focusing separately on in-degree and out-degree. A notable observation is that the maximum out-degree values are generally higher than those of in-degree. This suggests that a few prolific users, who frequently mention others, significantly influence the overall

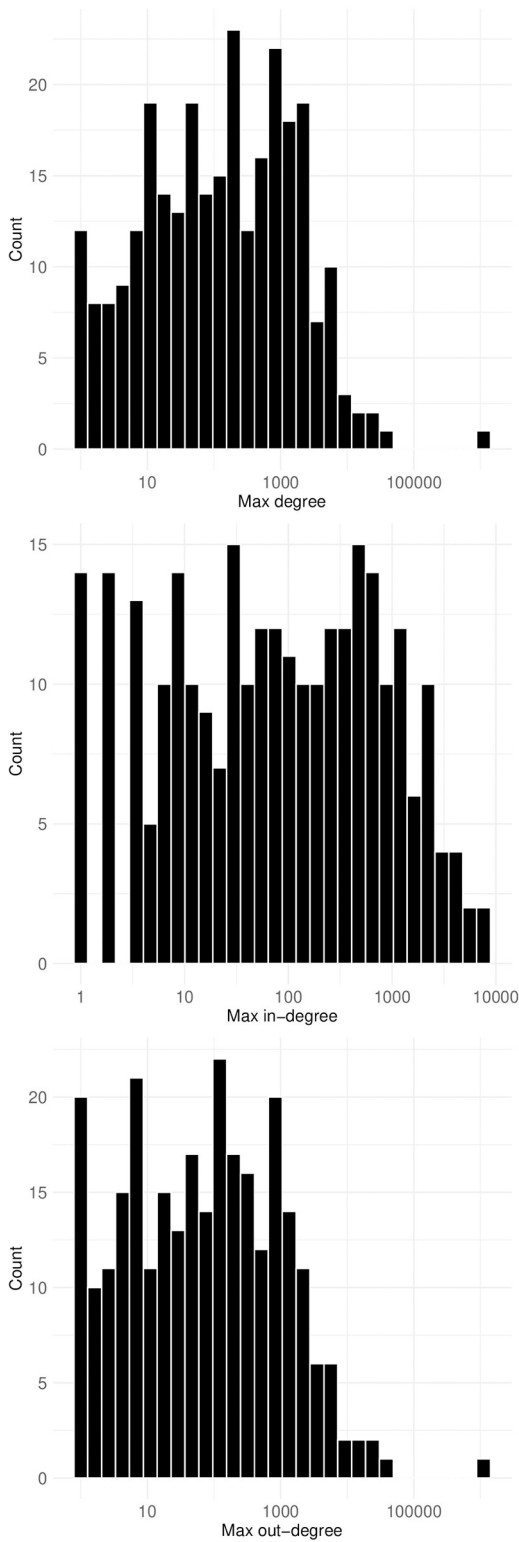

**Fig 16. Distribution of max node degrees, with the x-axis log-transformed while labels show the raw values.**
Notably, maximum out-degree values (bottom) surpass in-degree values (middle), indicating that a few prolific users drive the overall degree distribution. The mean maximum all degree (top) is 5130.5, with a median of 125, highlighting the significant influence of highly active users, such as channel owners and moderators.

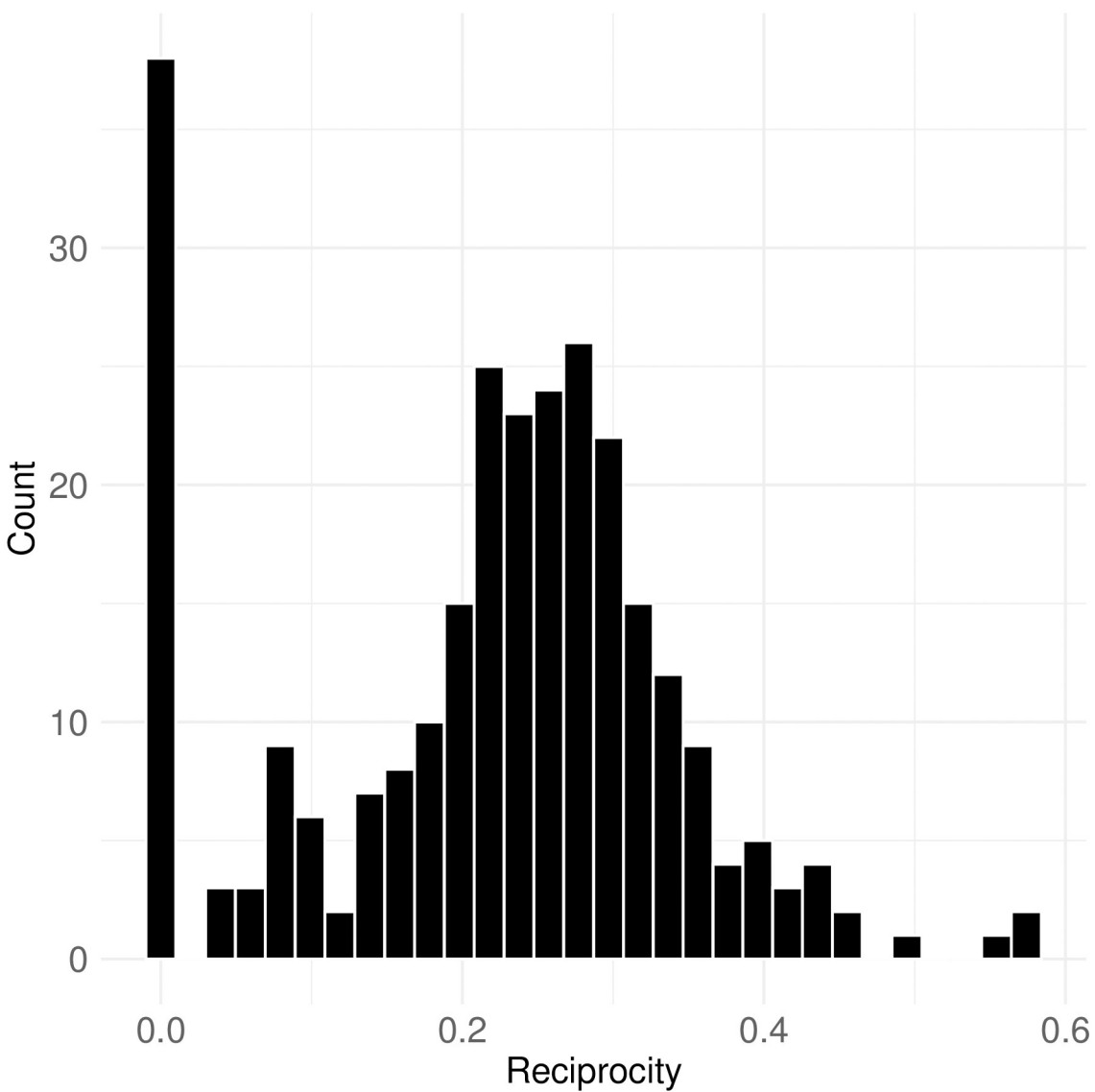

**Fig 17. Distribution of reciprocity scores, representing the proportion of reciprocal ties.**

degree distribution. Specifically, the peak values in the maximum all-degree histogram (top) align closely with those in the maximum out-degree histogram (bottom), indicating their substantial impact.

The mean value of the maximum all degree is 5130.5, with a median of 125, indicating significant skewness due to a few exceptionally active users, as evident from the histograms. These users, often channel owners or moderators, play roles such as answering questions during streams or enforcing streaming rules, contributing extensively to the network's edges. They can be considered opinion leaders within these reference networks, owing to their disproportionately large number of connections.

The reciprocity scores of all reference networks are depicted in Fig 17. The reciprocity score represents the proportion of reciprocal ties within each network. The mean reciprocity score is 0.22, indicating that, on average, 22% of connections among nodes are reciprocal. This means

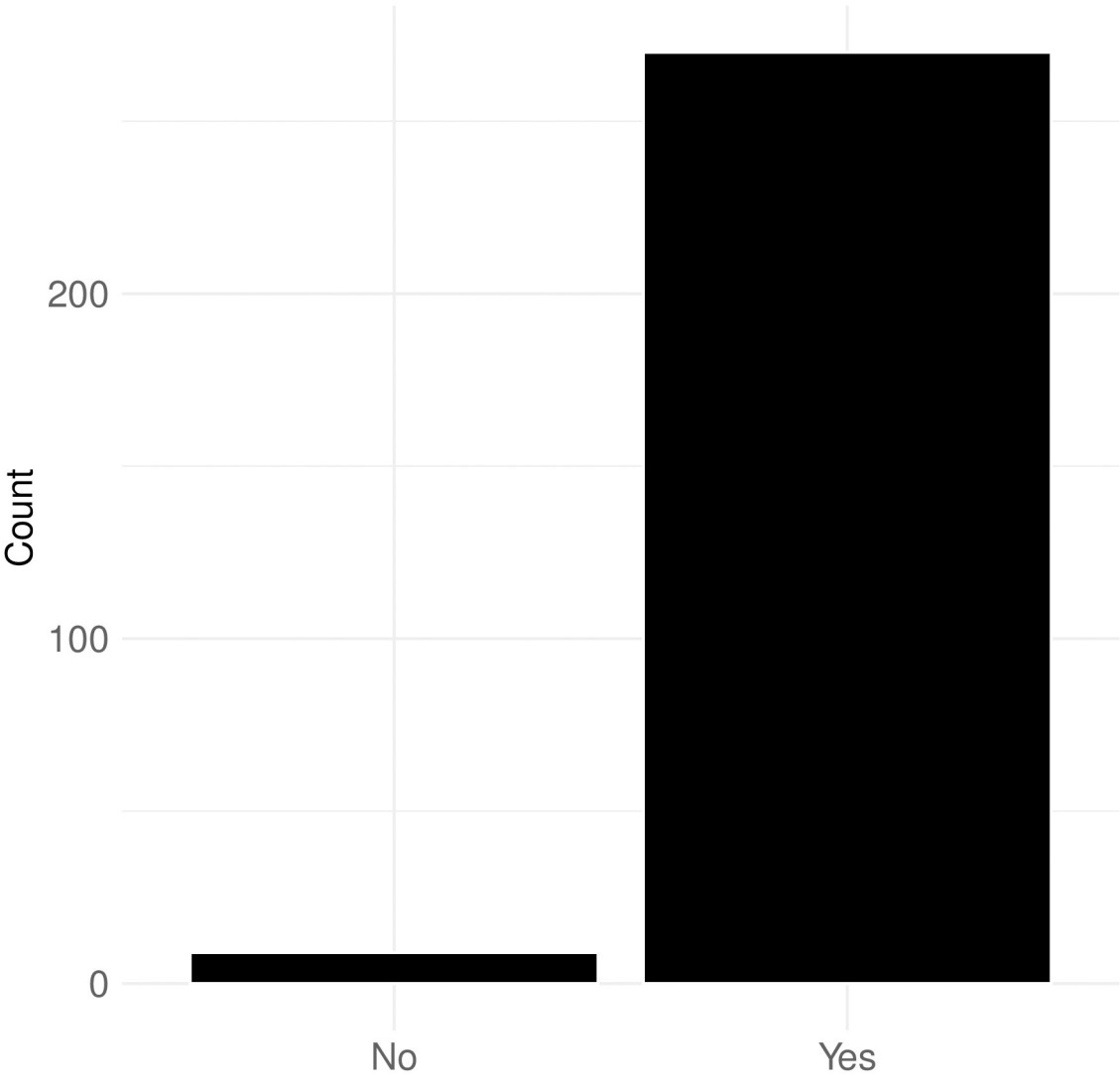

**Fig 18. Does network shows the power-law degree distribution?**

approximately one-fifth of interactions within each reference network involve mutual mentions between users. This finding aligns with Fig 16, which highlights the dominance of some highly active users in the communication network, where lower reciprocity is expected.

Fig 18 demonstrates that many mention networks follow a power-law degree distribution. I fitted a power-law degree distribution to each network and used the Kolmogorov-Smirnov test to determine if we could reject the null hypothesis—that the network could be derived from this distribution. The results show that over 96% of the networks conform to the fitted power-law degree distribution, similar to those observed on other social media platforms [46–48]. This implies that the structural characteristics of mention networks in political streaming chats on Twitch are not significantly different from those on other social media platforms. In other words, despite the technological differences inherent in real-time interaction and streaming chat, the fundamental structure of political communication networks on Twitch remains similar to those on other platforms.

## 9 Conclusion

In this article, I have addressed three main questions regarding Twitch politics: 1) Who are the political Twitch streamers? 2) What types of political content are discussed during streams? 3) How do audiences of political streams interact with each other? These questions were explored by focusing on the streaming chat function, which is central to the platform's communication technologies.

To answer the first question, I utilized the Twitch API and supervised machine-learning techniques to identify 574 political streamers. This study offers an important initial attempt to identify political actors on the platform, providing a valuable methodology for future research, though it is limited by its focus on a specific timeframe and does not fully account for the role of incidental political content from apolitical streamers.

For the second question, I employed topic modeling to examine the content of political streams. Since the content of political streams on Twitch was previously unknown, this study contributes by offering informative snapshots of this content. Notably, I found that identity-related topics are frequently discussed, highlighting how real-time interaction technology may influence topic choices among political actors. However, as the timeframe of the data collection coincided with significant global events, such as the Russian invasion of Ukraine, the prevalence of certain topics is likely influenced by these circumstances. This highlights the need for extended analyses across broader periods to capture the evolving nature of political discourse on Twitch.

To address the third question, I created and analyzed user-reference networks within each political streamer's chatroom. The analysis revealed that a small number of audience members dominate the communication network by frequently referring to one another. Additionally, despite the technological differences of real-time interaction and streaming chat, the fundamental structure of political communication networks on Twitch resembles those on other platforms. Most user-reference networks follow a power-law distribution, similar to communication networks on other social media platforms [46–48].

While this study offers valuable insights into political communication on Twitch, several limitations must be addressed. This study only analyzed text data from stream titles and profiles of streamers who used three specific game names. There may be political streamers who discuss political issues without explicitly indicating so in their titles or profiles, limiting the comprehensiveness of political streamer identification. Political discourse in apolitical streams are also not captured. Additionally, the analysis is based on a specific timeframe, and a broader timeframe may reveal different patterns of political content and engagement. Further research could explore incidental political content, partisanship, or toxicity within chat posts, offering more detailed insights into political communication on Twitch. These insights could contribute to the literature on both partisanship and political communication.

## Supporting information

**S1 Appendix. More information on data collection, coding rule, and the results of topic models.**
(PDF)

## Acknowledgments

I am grateful to Bruce Desmarais and Kevin Munger for help with various comments and guidance from the beginning of the project. I want to also thank Taegyoon Kim, Byunghwee Lee, Cyanne E. Loyle, Charles Seguin, and Tiago Ventura for helpful comments and

discussions. Special thanks to one of the authors of Flores et al. (2019), Joseph Seering, it was possible to successfully go through the data collection process all because of his help. Streaming chat posts collection was largely done based on the codes in his Github: https://github.com/josephseering/twitchtext, all the credits related to chat post scraper should be directed to the piece.

## Author Contributions

**Conceptualization:** Sangyeon Kim.

**Data curation:** Sangyeon Kim.

**Formal analysis:** Sangyeon Kim.

**Investigation:** Sangyeon Kim.

**Methodology:** Sangyeon Kim.

**Visualization:** Sangyeon Kim.

**Writing – original draft:** Sangyeon Kim.

**Writing – review & editing:** Sangyeon Kim.

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
