## [Decision Letter · Decision Letter 0]

1 Sep 2024

PONE-D-24-27781Understanding Political Communication and Political Communicators on TwitchPLOS ONE

Dear Dr. Kim,

Thank you for submitting your manuscript to PLOS ONE. After careful consideration, we feel that it has merit but does not fully meet PLOS ONE’s publication criteria as it currently stands. Therefore, we invite you to submit a revised version of the manuscript that addresses the points raised during the review process.

As also mentioned by the reviewers, all the conclusions drawn in the paper should be clearly supported by the data. Additionally, the paper should facilitate the reproducibility of the results by providing easy access to the data that has been used in the context of this study.

We look forward to receiving your revised manuscript.

Kind regards,

Liviu-Adrian Cotfas

Academic Editor

PLOS ONE

3. In your Methods section, please include additional information about your dataset and ensure that you have included a statement specifying whether the collection and analysis method complied with the terms and conditions for the source of the data.

5. We note that Figures 1,12 and A1 in your submission contain copyrighted images. All PLOS content is published under the Creative Commons Attribution License (CC BY 4.0), which means that the manuscript, images, and Supporting Information files will be freely available online, and any third party is permitted to access, download, copy, distribute, and use these materials in any way, even commercially, with proper attribution. For more information, see our copyright guidelines: http://journals.plos.org/plosone/s/licenses-and-copyright.

a. You may seek permission from the original copyright holder of Figures 1,12 and A1 to publish the content specifically under the CC BY 4.0 license.

6. We notice that your supplementary figures and tables are included in the manuscript file. Please remove them and upload them with the file type 'Supporting Information'. Please ensure that each Supporting Information file has a legend listed in the manuscript after the references list.

Reviewers' comments:

Reviewer's Responses to Questions

**Comments to the Author**

1. Is the manuscript technically sound, and do the data support the conclusions?

Reviewer #1: Yes

Reviewer #2: Partly

2. Has the statistical analysis been performed appropriately and rigorously? 

Reviewer #1: Yes

Reviewer #2: No

3. Have the authors made all data underlying the findings in their manuscript fully available?

Reviewer #1: No

Reviewer #2: No

4. Is the manuscript presented in an intelligible fashion and written in standard English?

Reviewer #1: Yes

Reviewer #2: Yes

5. Review Comments to the Author

Reviewer #1: The articles investigates the use of Twitch for political communication. The author provides a very detailed account of the methodology used (data collection via Twitch APIs, supervised classification, topic modeling, network analysis), through which he identifies a large set of political streams, classifies the recurring topics and analyses the interactions between the users. While very interesting quantitative results are provided, the discussion of these results could sometimes be deepened and more connected with the (extensive) literature review of the first sections. While some more methodological details are given in the Supporting information section, no link is provided to the raw data used in the study.

Here follows a list of more precise recommendations :

- Page 5, "live streaming on Twitch is even more efficient because the time spent for streaming is the same as the length of the show" -> the formulation is not very clear ; it also implies that streamers are not spending time preparing their performance before the stream.

- Page 5, "Although not all streamers can earn a stable income -> It would be more fair to say that only a tiny minority of streamers can earn a stable income (data is hard to find, but it cannot exceed 1% of the streamers).

- Page 8, to properly answer the first question (Who are the political Twitch streamers ?), it would have been interesting to provide some insights on the identified profiles (based on the data actually collected through the API : registration date ? number of followers ? location ? age ? gender ?)

- Some figure captions are a bit short (ex : fig 2, fig 6, fig 11) and could be expended to guide the interpretation of the diagrams.

- Page 10, the "maximum chat frequency" indicator could be described more clearly : does it represent the activity of the most active user of each chat ?

- Page 12, the predominance of the "nazi" topic in International issues raises questions : is the use of nazi-associated words always linked to international politics-discussions, or could it also cover other political (or even non-political) uses as a derogatory term or an insult ?

- Fig 12, it would be interesting to add the Potchamp emote, since its usage is investigated in the previous figure.

Reviewer #2: I thank the authors for their submission and for engaging with the important questions of political engagement and communication on platforms outside of Twitter and YouTube.

In its current form, while the manuscript is interesting, it has several weaknesses that preclude me from recommending publication at this time.

First, the paper claims to provide a "comprehensive" description of political content and actors on Twitch ("This study is the first comprehensive attempt to identify political actors on the platform"), but this claim is almost certainly incorrect. The paper itself acknowledges (in the next paragraph) that it provides "informative snapshots", and the period of observation is both unclear and constrained. Twitch has been around since 2011, but this analysis appears constrained to 2021 and early 2022. Engagement with or acknowledgement of this limited timeframe is largely absent from the article. Likewise, the paper's label as comprehensive is itself undermined by the important point it makes about incidental exposure, wherein apolitical streamers occasionally leak their political perspectives. Studies of incidental exposure, such as Dreston et al., 2023, show this exposure is formative, so discussion of overtly political actors on a gaming platform is almost certainly incomplete without a deeper engagement of this topic. Certainly, identifying incidental political content on Twitch may not be straightforward (e.g., is something like the "Thanks, Obama" meme political?), but it is a major omission in the descriptive work of this article. Additional care should be taken to resolve this omission and the tension between the paper's claims of comprehensiveness and the role of apolitical streamers in political communication on Twitch.

Second, the datasets and collection process used to support the paper's analysis and conclusions is largely unclear to me. It seems that the political classifier was developed on a very small slice of data, then the content analysis was done on a different dataset collected several months later. And in both cases, the collection window seems quite small. Developing a political classifier from one week of data and then drawing strong conclusions from about 4 months of behavior on a platform that's over ten years old is problematic, especially as this timeframe includes the beginning of the large-scale Russian invasion in Ukraine. As I see it, the core contribution of this work is a description of political content on Twitch; that's a laudable goal, but the constrained and limited detail on data collection impedes this contribution. Perhaps a model paper from a description-focused venue like JQD would be useful, such as Brown et al., 2024 [2].

Third, the content analysis piece of this work is underdeveloped given its centrality to the work's contributions. Multiple resources exist on performing content analysis in social media, with Rodriguez and Storer, 2019 using a topic-modeling based approach similar to this paper [3]. Other resources [4-5] provide guidance on qualitative coding for this kind of content, and Wojcieszak et al., 2022 provides an excellent model for studying political discussion in online spaces. This paper's single-coder approach without a clear codebook guiding the analysis is problematic from a reproducibility standpoint, and it introduces problems around the boundaries of the topics, such as how identity politics differs from US politics. Without a clearer and more rigorous approach to this content characterization, it would be difficult to make claims about the kinds of political content popular on this platform.

Lastly, the statistical analyses of the distributions appears largely qualitative and unclear. For example, the paper claims the skewness associated with the distributions in Figure 5 and elsewhere, but I don't see quantitative evidence of this skew. The descriptions of the distributions also tend to state that the figures are in log scale, but the y axes don't tend to reflect that.

Ultimately, I think the paper is a good start to a contribution that would be a good fit for this venue or a descriptive space like JQD. In its current form though, the conclusions tend to be too strong given the underlying data and its constraints. If the authors could push more on extending the collection and strengthen their approach to content analysis, I think a nice contribution could be made.

## Other comments

- Generational dynamics have consistently been salient in politics, so the paper's claim that "As generational dynamics have been salient in recent American politics and other countries" seems problematic. Likewise, the paper's characterization that spread of political speech into streaming media and apolitical spaces is new is similarly problematic, as politics from apolitical celebrities or spaces is not new. I think these points speak to a larger issue about limited engagement with the political science literature.

- I don't understand the network structure in section 7

- Why do we have 279 networks for 478 streamers? Do some streamers not have netowrk connections in their chats?

- The ratio of isolates seems low for a platform that is mostly geared toward audience  streamer engagement

- Paragraphs on page 15 seem redundant

- I am unconvinced about the high computational costs of topic modeling. Where does the 5% sampling rate come from? Why not 6% or 2%? Need justification here

## Related Work

[1] Dreston JH, Neubaum G. How incidental and intentional news exposure in social media relate to political knowledge and voting intentions. Front Psychol. 2023 Dec 21;14:1250051. doi: 10.3389/fpsyg.2023.1250051. PMID: 38187440; PMCID: PMC10768061.

[2] Brown, M., Sanderson, Z., Graham, S., Kim, M., Tucker, J., & Messing, S. (2024). Digital town square? Nextdoor’s offline contexts and online discourse. Journal of Quantitative Description: Digital Media , 4. https://doi.org/10.51685/jqd.2024.icwsm.2

[3] Rodriguez, M. Y., & Storer, H. (2019). A computational social science perspective on qualitative data exploration: Using topic models for the descriptive analysis of social media data*. Journal of Technology in Human Services, 38(1), 54–86. https://doi.org/10.1080/15228835.2019.1616350

[4] Caplan, M. A., & Purser, G. (2019). Qualitative inquiry using social media: A field-tested example. Qualitative Social Work, 18(3), 417-435. https://doi.org/10.1177/1473325017725802

[5] Sloan, Luke, and Anabel Quan-Haase. "The SAGE handbook of social media research methods." (2022): 1-100.

[6] Magdalena Wojcieszak et al. ,Most users do not follow political elites on Twitter; those who do show overwhelming preferences for ideological congruity.Sci. Adv.8,eabn9418(2022).DOI:10.1126/sciadv.abn9418

6. PLOS authors have the option to publish the peer review history of their article (what does this mean?). If published, this will include your full peer review and any attached files.

Reviewer #1: No

Reviewer #2: No

---

## [Author Response · Author response to Decision Letter 0]

17 Oct 2024

Dear Editor and Reviewers,

Thanks for giving me a chance to revise the manuscript. I have attached the separate file "Response to Reviewers" that addresses that comments raised during the review process.

Best,

Sangyeon

---

## [Decision Letter · Decision Letter 1]

11 Nov 2024

Understanding Political Communication and Political Communicators on Twitch

PONE-D-24-27781R1

Dear Dr. Kim,

We’re pleased to inform you that your manuscript has been judged scientifically suitable for publication and will be formally accepted for publication once it meets all outstanding technical requirements.

Kind regards,

Liviu-Adrian Cotfas

Academic Editor

PLOS ONE

Additional Editor Comments (optional):

Reviewers' comments:

Reviewer's Responses to Questions

**Comments to the Author**

1. If the authors have adequately addressed your comments raised in a previous round of review and you feel that this manuscript is now acceptable for publication, you may indicate that here to bypass the “Comments to the Author” section, enter your conflict of interest statement in the “Confidential to Editor” section, and submit your "Accept" recommendation.

Reviewer #1: All comments have been addressed

Reviewer #3: All comments have been addressed

2. Is the manuscript technically sound, and do the data support the conclusions?

Reviewer #1: Yes

Reviewer #3: Yes

3. Has the statistical analysis been performed appropriately and rigorously? 

Reviewer #1: Yes

Reviewer #3: Yes

4. Have the authors made all data underlying the findings in their manuscript fully available?

Reviewer #1: Yes

Reviewer #3: Yes

5. Is the manuscript presented in an intelligible fashion and written in standard English?

Reviewer #1: Yes

Reviewer #3: Yes

6. Review Comments to the Author

Reviewer #1: (No Response)

Reviewer #3: The manuscript, 'Understanding Political Communication and Political Communicators on Twitch', has successfully addressed all the prior reviewers' feedback. I support its publication in its current version.

7. PLOS authors have the option to publish the peer review history of their article (what does this mean?). If published, this will include your full peer review and any attached files.

Reviewer #1: No

Reviewer #3: No

---

## [Editor Report · Acceptance letter]

15 Nov 2024

PONE-D-24-27781R1 

PLOS ONE

Dear Dr. Kim, 

I'm pleased to inform you that your manuscript has been deemed suitable for publication in PLOS ONE. Congratulations! Your manuscript is now being handed over to our production team.

Kind regards, 

on behalf of

Dr. Liviu-Adrian Cotfas 

Academic Editor

PLOS ONE